

# Dramatic decline and limited recovery of a green crab (*Carcinus maenas*) population in the Minas Basin, Canada after the summer of 2013

Brady K. Quinn

Department of Biological Sciences, University of New Brunswick, Saint John, New Brunswick, Canada

## ABSTRACT

This paper reports the results of a ten-year monitoring program of an Atlantic Canadian population of green crabs, *Carcinus maenas*, in the Minas Basin of the Bay of Fundy. Intertidal densities, sex and reproductive ratios, juvenile recruitment, subtidal catch-per-unit-effort (CPUE), and sizes of crabs in this population were recorded from 2008 to 2017. In 2013 intertidal densities, mean crab sizes, subtidal CPUE, and proportions of crabs mature and reproducing all dramatically decreased to all-time lows, and large crabs virtually disappeared from the population. From 2014 to 2017 the population partially recovered but remained in an altered state. Potential causes of interannual changes to this population were investigated by correlating intertidal densities to 257 monthly environmental variables and performing stepwise multiple regression analyses. Crab densities in a given year were best explained by potential settlement during the summer and the maximum sea-surface temperature during March of the same year. However, potential roles of other factors (e.g., autumn winds, summer temperatures, North Atlantic Oscillation index) could not be ruled out. Changes in abundances of other species in the area, particularly predators and prey of green crabs, have also been observed and present possible alternative causative agents that should be investigated. Populations of other marine species in the Gulf of Maine-Bay of Fundy region within which the Minas Basin is situated have also been reported to have undergone dramatic changes in and after 2013, suggesting the occurrence of some oceanographic event or regime shift in the region. Declines to the monitored crab population in this study may have resulted from this same 2013 event. These observations have implications for recruitment to marine populations in this region.

# INTRODUCTION

Detailed observations over time made on organisms of the same species in one or more selected locations, constituting the same putative 'population(s)', is a staple of studies in marine biology and ecology in general (*Bertness et al., 1992*; *Santos & Simon, 1980*; *McGaw, Edgell & Kaiser, 2011*; *Palumbi & Pinsky, 2014*). Such monitoring can provide essential information about the life history and demography of the species of interest

Corresponding author
Brady K. Quinn, bk.quinn@unb.ca

(*Hoskin et al., 2011*), its interactions with other species in the same community (*Seitz, Knick & Westphal, 2011*), and abiotic or biotic factors influencing recruitment to its populations (*Scrosati & Ellrich, 2016*). When observational periods coincide with particularly disruptive events, for example natural disasters (*Sato & Chiba, 2016*), the introduction of new invasive species (*Delaney et al., 2008*), or climate-driven shifts in oceanographic regimes (*Mills et al., 2013*; *Pinsky et al., 2013*), the impacts of such events on the studied species can be illuminating. Changes in abundances and the size or reproductive structure of species' populations following such events, as well as overall shifts in community specific composition, illustrate which ecological changes impact particular species and how (*Ruth & Berghahn, 1989*; *Palumbi & Pinsky, 2014*; *Bertness et al., 1992*). These observations provide information that can potentially be used to forecast longer-term impacts of such changes into the future, which is essential for management and conservation of marine species (*Ruth & Berghahn, 1989*; *Pinsky et al., 2013*; *Wahle & Carloni, 2017*).

When a non-native species is introduced to a new location its impacts on native biota can be substantial (e.g., *Klassen & Locke, 2007*; *Scalici & Gherardi, 2007*), so monitoring introductions and their impacts provides important ecological information. Once an introduced species establishes an invasive population, however, it takes on a distinct role within the invaded community and is essentially a part of the invaded ecology (e.g., *Boudreau & Hamilton, 2012*). Invaders may, however, be more sensitive to environmental perturbations in their invasive range than are native biota because invaders, unlike native biota, do not have the same evolutionary history with environmental conditions in their invaded range as do natives (*Grosholz et al., 2000*; *Kienzle, 2015*; *MacDonald et al., 2018*). It is thus possible that ecological shifts potentially affecting an entire biological community may first impact an invasive species before having similar effects on native taxa. Monitoring invasive as well as native populations can thus potentially serve as an early warning system for changes to the community or its ecosystem.

The Bay of Fundy is an inlet of the Gulf of Maine, located along the Atlantic coast of eastern North America, with extremely large tidal ranges (up to 16.8 m) and distinct marine communities, particularly in the Minas Basin and Chignecto Bay portions of the upper Bay (*Daborn & Pennachetti, 1979*; *Parker, Westhead & Service, 2007*). Marine communities within the Gulf of Maine-Bay of Fundy system are currently facing a number of sources of stress and change, including rapid warming of ocean waters (*Mills et al., 2013*; *Pinsky et al., 2013*) and introduction of several invasive species (*Moore et al., 2014*; *Klassen & Locke, 2007*). Particularly influential in recent years have been very warm summers, such as the 2012 'heat wave', and possible associated shifts in oceanographic circulation (*Mills et al., 2013*). Indeed, many species in this system demonstrated low abundances or recruitment in the year 2013 (e.g., *Clements, 2016*; *Wahle & Carloni, 2017*; see also Discussion), signalling some major disruption in the system that bears further investigation. One invasive species that has become well-established within the Bay of Fundy is the European green shore crab, *Carcinus maenas*. This species reached the Bay of Fundy in the 1950s and became established and numerous within the upper bay (i.e., the Minas Basin) in the late 1990s (*Klassen & Locke, 2007*). Green crabs are aggressive consumers of a wide range of prey species as predators and

scavengers (*Crothers, 1968*; *Trussell, Ewanchuck & Bertness, 2003*; *Klassen & Locke, 2007*; *Boudreau & Hamilton, 2012*) and compete with native species preying on similar food sources (*Grosholz et al., 2000*; *Haarr & Rochette, 2012*). This species thus plays significant roles within food webs of marine communities it has invaded (*Wong & Dowd, 2015*).

This paper describes the results of efforts to monitor an established invasive population of green crabs (Fig. 1A) at Clarke Head, Nova Scotia (NS), Canada (Figs. 1B, 1C), within the Minas Basin of the upper Bay of Fundy (Fig. 1D). Green crabs first appeared at this site ca. 1999, and very quickly increased in abundance and almost completely displaced local rock crabs (*Cancer irroratus*) from intertidal environments at Clarke Head by 2002 (BK Quinn, pers. obs., 2000–2018). A specific monitoring protocol was established by the author in 2008 and continued thereafter. Monitoring thus unfortunately missed the early years in which green crabs first appeared and became abundant at this location. However, this relatively new population within the area was still monitored to track its health and dynamics, as doing so is potentially useful to forecast impacts of changes to invader numbers on native biota, and to use it as a potential early warning system for changes to the marine ecology within the study area. During the monitoring period (2008–2017), a substantial change to the green crab population was observed beginning in 2013 and persisting to the present. Within the present paper, the characteristics of the crab population before, during, and after this change are examined, and environmental changes potentially responsible are compared with changes in the crab population. Observations by recreational fishers and locals within the study area of biotic changes to crab bycatch and potential predators and prey of green crabs are also considered as a form of Local Ecological Knowledge (LEK; *Cosham, Beazley & McCarthy, 2016*) providing further avenues for research into causes of crab population changes. Findings for green crabs in the present study are also related to concurrent changes in the study region from 2013 onwards reported for other species and their potential impacts.

## MATERIAL AND METHODS

### Study site and background

This study was carried out along the intertidal and subtidal zones off of Clarke Head, a point of land near Parrsboro, NS, Canada, tipped by a small cove surrounded by cliffs of sedimentary rock (Figs. 1B, 1C). Clarke Head is located on the north shore of the Minas Basin, an inlet of the upper Bay of Fundy (Fig. 1D). The upper intertidal zone at Clarke Head consists of a steeply sloping gravel beach (Figs. 1B, 1C). The middle intertidal is a relatively flat area composed mainly of sand or a mud-sand mix, interrupted by a series of rocky reefs of the same composition as the surrounding cliffs and covered primarily with rockweed, *Ascophyllum nodosum* (Figs. 1B, 1C). A previous study (*Quinn, 2016*) reported observations at the boundary between the upper and middle intertidal zones at this location. A large middle intertidal mud flat is located to the north of this area, just outside of the cove (Fig. 1C), and receives considerable freshwater input via a stream called Swan Creek. The lower intertidal off of Clarke Head is defined by a large series of gravel and sand bars to the south and north (Figs. 1B, 1C), in places bearing large aggregations

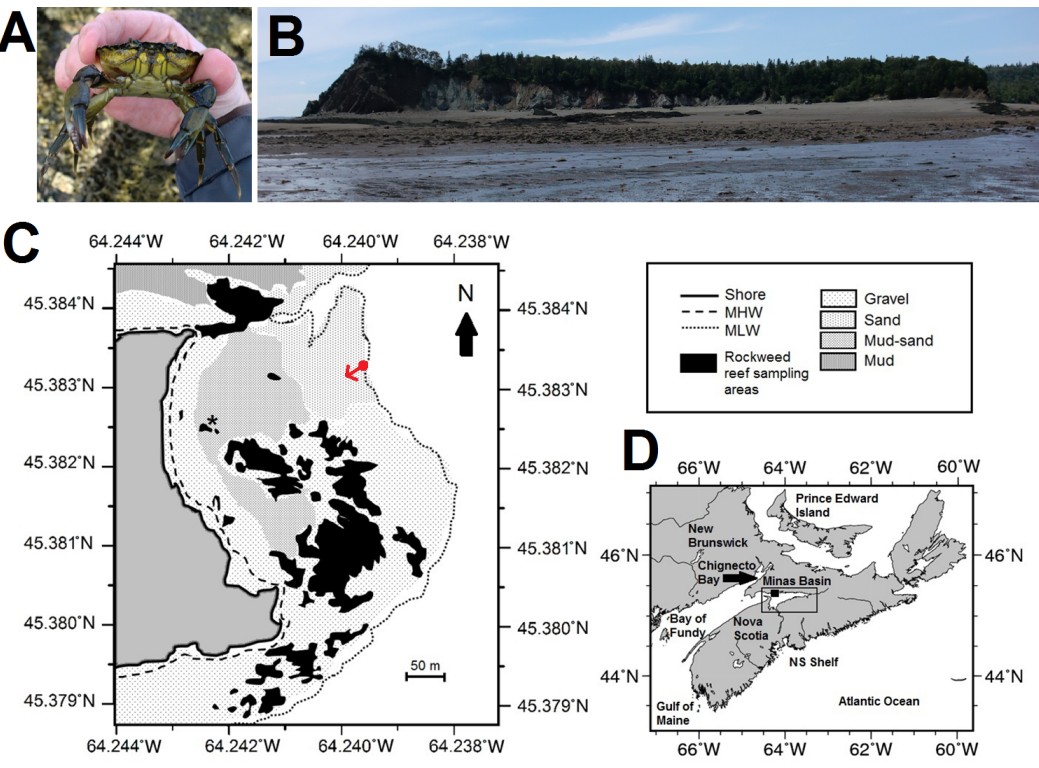

**Figure 1 Map, features, and location of the study site at Clarke Head, Nova Scotia (NS).** (A) Photograph of an adult male green crab (*C. maenas*) examined at the study site. (B) Photograph of the study site, taken toward the shore from the low tide mark from the approximate location of the red dot in (C); reefs surveyed for crabs are visible mainly on the right and left sides of the image. (C) Map showing details of the study site, including bottom substrate types on the intertidal zone; solid gray area is land, white areas are subtidal, black areas are rockweed reefs where crabs were sampled in this study, asterisk is the location of a previous study (*Quinn, 2016*). (D) Map showing the location of the study site (solid square) and Minas Basin (rectangular outline) within Atlantic Canada. Photographs in (A) and (B) were both taken by the author in July 2015, and both maps were made by the author (C) by hand or (D) using Generic Mapping Tools GMT-5.3.1 (*Wessel et al., 2013*; available online from http://gmt.soest.hawaii.edu/).

of barnacles (*Semibalanus balanoides*) and blue mussels (*Mytilus* sp.). Oceanic circulation within the cove generally runs from southeast to northwest on flood tides and reverses on ebb tides. Extremely low intertidal to high subtidal areas off of Clarke Head are also occasionally exposed on spring tides, and consist of alternating sandy flats and rocky (cobble) ridges bearing seaweeds such as Irish moss (*Chondrus crispus*), sea lettuce (*Ulva* sp.), and dulse (*Palmaria palmata*). These features are also interspersed with kelp beds (*Saccharina latissima*) and patches of sea grass (*Zostera marina*). At low tide an expanse of seafloor extending up to 2.5 km from shore is exposed at this site. The features of this site (patches of rocky intertidal interspersed among sand and mud flats) are typical of the Minas Basin region (*Daborn & Pennachetti, 1979*; *Parker, Westhead & Service, 2007*), although the rocky areas here are somewhat more extensive than is typical. The extent and accessibility of the seafloor off of Clarke Head thus presents opportunities for considerable observation and study of a variety of marine life.

**Table 1 Sampling effort during intertidal and subtidal green crab monitoring at Clarke Head, NS, in different years and months.** Columns for intertidal crab sampling in June–September list the number of sampling trips made per each month. Quadrats used measured 1 m$^2$.

| Year | Intertidal crab sampling | | | | | | Subtidal crab bycatch | |
|---|---|---|---|---|---|---|---|---|
| | June | July | August | September | Total # quadrats sampled | Total # crabs found | Total # fishing trips | Total # crabs caught |
| 2008 | 1 | 2 | 2 | 1 | 140 | 394 | 8 | 35 |
| 2009 | 2 | 3 | 3 | 3 | 220 | 896 | 5 | 18 |
| 2010 | 1 | 1 | 1 | 1 | 80 | 641 | 10 | 39 |
| 2011 | 1 | 0 | 0 | 1 | 40 | 198 | 8 | 29 |
| 2012 | 0 | 2 | 3 | 2 | 140 | 1,232 | 6 | 32 |
| 2013 | 1 | 5 | 6 | 3 | 320 | 86 | 9 | 1 |
| 2014 | 3 | 4 | 4 | 2 | 240 | 112 | 5 | 3 |
| 2015 | 1 | 2 | 1 | 1 | 80 | 168 | 4 | 7 |
| 2016 | 0 | 1 | 0 | 1 | 30 | 103 | 1 | 2 |
| 2017 | 0 | 1 | 2 | 0 | 88 | 210 | 3 | 8 |

## Green crab biological information

Data collection during monitoring and subsequent analyses were informed (i.e., the data partitioned and interpreted) based on information from the literature about green crab biology (*Crothers, 1967*; *Crothers, 1968*; *Berrill, 1982*; *Klassen & Locke, 2007*). Among green crabs, the sexes can be reliably distinguished at carapace width (CW) $\geq$ 15 mm (*Crothers, 1967*; *Crothers, 1968*), which corresponds to an age of 1–2 years. Female crabs become ovigerous at CW $\geq$ 30–40 mm depending on region (*Klassen & Locke, 2007*). Maximum size for female crabs is ~75 mm CW whereas males can exceed 90 mm CW, and maximum crab lifespan is thought to be 3–7 years depending on region (ca. 6 years in Maine; *Berrill, 1982*). In eastern North America, ovigerous females are generally present in the summer, between June–October, and then larval release occurs several weeks to months later in late summer or early fall (August–December, peak in September). Once released, larvae develop through five planktonic larval stages (four zoeae and one decapodid, commonly called a megalopa), lasting from ca. 24 to 82 days depending on water temperature (*Dawirs, 1985*). The last larval stage then settles to the seabed between August and October (*Berrill, 1982*; *Klassen & Locke, 2007*).

## Sampling methodology

Beginning in 2008, opportunistic sampling trips to Clarke Head were regularly undertaken up to 15 times per year over the course of the summer (June-September) during the day at low tide (Table 1). Preliminary surveys of the study area found almost no green crabs on sand, mud, or mud-sand mix flats or in gravel areas during daytime low tides, so sampling was confined to intertidal rockweed-covered reefs (black areas in Fig. 1C), which were the most suitable intertidal habitats for crabs in the area and allowed for the most straightforward and repeatable sampling protocol. Sampling was confined to the summer months because this is the time of the year when crabs are most abundant on the intertidal

and when most events of interest in their life history take place (see above); also, crabs are largely absent from the intertidal in this region during the winter.

During sampling trips, as many 1 m$^2$ quadrats as tide and time allowed were haphazardly sampled along all intertidal rockweed reefs located between approximately 7–10 m above chart datum at the site (see Table 1 for annual and monthly sampling effort and Figs. 1B, 1C for the distribution of reefs sampled). If they fell within a quadrat, any rocks were flipped and any crevasses or rockweed were thoroughly searched. The number of green crabs present in the quadrat was recorded, as was the size (CW $\pm$ 1 mm) and sex of all crabs found. Sex could be determined for crabs with CW of 15 mm or greater based on the shape of the abdomen, which is much broader in females than in males (*Crothers, 1968*). Any crabs found that were smaller than 15 mm CW were classified as juveniles. Whether females were ovigerous (carrying eggs beneath their abdomen) was also recorded. This allowed intertidal densities of crabs (number per m$^2$) to be calculated, as well as information on crab population dynamics such as mean sizes, sex ratios, ovigerous female proportions, and juvenile abundances.

Importantly, relatively large green crabs retreat on the ebb tide and remain in subtidal areas, rather than hiding on the intertidal (*Hunter & Naylor, 1993*). As such, intertidal surveys could underestimate crab abundances on their own. Many traditional and standard methods for sampling crab populations in both subtidal and intertidal areas exist, such as the use of Fukui traps in subtidal areas (*MacDonald et al., 2018*) or intertidal pitfall traps that capture crabs that enter intertidal areas at low tide (*Kent & McGuinness, 2006*). However, these methods require some specialized materials to build and expertise to use and maintain. The intertidal sampling program in the present study was established with no additional funding support and in such a way that non-scientist volunteers could readily assist in sampling. As such, sampling methods were initially confined to the daytime intertidal and kept as simple as possible (i.e., the use of quadrats), and in later years this approach was maintained for consistency. While this approach does have shortcomings (see Discussion), it allowed for a long time series that reasonably captured interannual changes in the crab population to be obtained. Nonetheless, to control for the potential issue of intertidal sampling missing larger, subtidal crabs, unintentional capture (i.e., bycatch) of large, subtidal green crabs (all $\geq$ 50 mm CW) during recreational flounder fishing trips was recorded. During a typical fishing trip (in years when the crab population was high), it was common for one to catch as many or more green crabs as flounder or other fish species (i.e., upwards of 5–10 or more crabs per hand-line; BK Quinn, pers. obs., 2000–2018), so crab bycatch during such trips likely provides a good index of their abundance in subtidal areas. Data from up to 10 fishing trips per year were collected by consulting with local recreational fishers (Table 1). It was not possible to obtained detailed information of crab sex or size from these consultations, as fishers did not generally make such detailed observations. The total number of green crabs caught per year was divided by the total number of fishing trips per year from which information was obtained (number of fishers $\times$ number of trips) to create an index of annual catch per unit effort (CPUE) of large subtidal crabs. While not equivalent to intertidal densities, this does give a similar index of large, subtidal crab abundance and how it changed among years.

## Statistical analyses

### Comparisons among years

Intertidal densities (number of individuals m$^{-2}$) of green crabs were calculated per each sampling trip as the total number of crabs found in all quadrats during that trip divided by the total number of 1 m$^2$ quadrats sampled (see Table 1). During preliminary analyses, it was found that densities did not differ significantly or in any consistent manner among sampling trips taken during different months within each year (ANOVA, $F_{3,35} = 1.679$, $p = 0.189$), so all sampling trips from the same year were pooled to make overall comparisons among years. Proportions of crabs mature (CW $\geq$ 15 mm), proportion of mature crabs female, proportion of female crabs ovigerous, total number of juveniles observed, and average size of crabs (overall and of different sexes) were also calculated based on all crabs in each year.

Overall intertidal crab densities were compared among sampling years using a one-way analysis of variance (ANOVA), in which the factor was year (treated as a categorical factor with ten levels). If a significant effect of year was found, post-hoc comparisons were made among years using Tukey's Honestly Significant Difference (HSD) test. Because the year 2013 showed extremely reduced densities compared to all other years (see Results and Table 1), especially the years preceding it (2009–2012), a further set of post-hoc comparisons of all other years against 2013 was carried out using Dunnet's pairwise multiple comparison $t$-test. All of the aforementioned analyses were carried out in IBM SPSS Statistics 23 (SPSS Inc., 2015). Overall mean values and 95% confidence intervals (C.I.s) were also calculated across the sampling period (2008–2017) of intertidal density and subtidal CPUE to evaluate where specific years fell in relation to these overall averages.

### Comparisons among biological characteristics of the crab population

As various biological characteristics of the green crab population at Clarke Head potentially affecting its growth and recruitment dynamics (e.g., sex and size structure) were also quantified in each year, whether the values of these characters in each year were correlated to changes in crab abundance was tested (Table 2). Specifically, Pearson's correlation coefficient ($r$) values and the associated $p$-value of the comparison of correlation coefficients against zero (*Cohen, West & Aiken, 2003*; https://www.danielsoper.com/statcalc/) were calculated between average intertidal densities calculated for each year ($N = 10$ pairs per comparison because there was only one value per year) and the following biological characteristics of the crab population in each year: proportion of all crabs that were mature (CW $\geq$ 15 mm), proportion of mature crabs that were female, proportion of females that were ovigerous, mean size of all crabs, mean size of all females, mean size of non-ovigerous females, mean ovigerous female size, mean male size, and total number of juvenile crabs observed. Correlation coefficients were also calculated between each of these biological predictor variables ($N = 10$ pairs) to determine whether they could be included together in later analyses. Whether annual subtidal CPUE of large crabs was significantly correlated with mean annual intertidal density was also determined to assess to what extent fluctuations in these values across years agreed.
**Table 2 Correlations between biological characteristics of the crab population in each sampled year and average observed intertidal crab densities.** Values presented in the table are Pearsons correlation coefficients ($r$). Significant correlations ($r < -0.630$ or $r > +0.630$, $N = 10$, $p \le 0.05$; *Cohen, West & Aiken, 2003*; https://www.danielsoper.com/statcalc/) are indicated by bold text.

| Predictor variable type | | Correlation |
|---|---|---|
| **Proportion of crabs** | Mature | $+0.486$[***] |
| | Female | $+0.407$[***] |
| | Ovigerous | $-0.506$[***] |
| **Mean size (mm CW)** | All crabs | $+0.544$[***] |
| | Female (all) | $+0.499$[***] |
| | Ovigerous female | $+0.549$[***] |
| | Female (non-ovigerous) | $+0.373$[**] |
| | Male | $+0.528$[***] |
| **Number juvenile crabs observed** | Total | $+0.443$[***] |

Notes.
[**] $0.001 < p \le 0.01$.
[***] $p < 0.001$.

### Potential causes of changes in the sampled crab population

To investigate potential causes of fluctuations in the green crab population at Clarke Head, a series of correlations, stepwise multiple regressions, and model selection analyses were carried out. Only intertidal densities were evaluated in this way because they provided a larger sample size (more sampling trips), and thus a more powerful analysis than could be achieved with subtidal CPUE data. This approach was unlikely to have missed important patterns because intertidal density and subtidal CPUE were strongly correlated (see 'Results').

As a first step, whether average annual intertidal densities were correlated with a number of environmental (abiotic) measures potentially impacting crab recruitment or survival was tested. Because conditions in different time periods might impact different phases of the crab life cycle (e.g., effects on larvae in summer, adult or juvenile survival over winter), environmental measures were averaged or otherwise calculated for each month of the year. Comparisons were made between intertidal density in a year and environmental conditions in the summer and autumn months (July–December) of the previous year and the winter, spring, and summer months (January–September) of the same year; months in other seasons were not considered because they were unlikely or unable to impact densities observed in the summer of a given year.

Air and water temperatures were considered because temperature in general has powerful impacts on the physiology, survival, growth, moulting, and larval development of crustaceans, including green crabs (*Byrne, 2011*; *Dawirs, 1985*; *Nagaraj, 1993*). Intertidal organisms like green crabs are affected by water temperature specifically during high tides, and by air temperature while exposed during low tides (*Bertness et al., 1992*; *Scrosati & Ellrich, 2018*). Precipitation, wind speed, and wind direction were considered because precipitation impacts freshwater runoff and coastal salinity, which potentially impacts crab physiology (*Nagaraj, 1993*; *Klassen & Locke, 2007*), winds potentially impact larval retention during certain seasons (*Bertness, Gaines & Wahle, 1996*), and both wind and precipitation

can indicate the frequency and severity of storms impacting survival and movement of all life stages (*Meehl et al., 2000*). Variability in abiotic features of the environment, such as temperature, can have distinct effects on organism performance from those of mean values alone (e.g., *Niehaus et al., 2012*), so not only average values of each environmental variable but also their variability were examined. The North Atlantic Oscillation (NAO) index is an index of the difference in atmospheric pressure between Iceland and the Azores, which is associated with large-scale climatic variability in the North Atlantic Ocean (*Hurrell et al., 2003*). Large negative values of the NAO index have been associated with cold and snowy conditions in eastern North America, including lower water temperatures and salinity in the Gulf of Maine-Scotian Shelf system, which includes the Bay of Fundy (*Petrie, 2007*). The NAO index may thus capture large-scale physical processes impacting marine life, so it was also examined. Finally, variations in oceanographic circulation from year to year can impact settlement by planktonic larvae of benthic invertebrates, and thus subsequent recruitment to their populations (*Bertness et al., 1992*). Potential settlement predicted by oceanographic dispersal models can indicate a role of oceanography in recruitment.

Most environmental data outlined above, including daily mean air temperature (in °C), variability (= standard deviation, or SD) of daily air temperature, minimum daily air temperature, maximum daily air temperature, total daily precipitation (in mm), variability (SD) of daily precipitation, maximum daily precipitation, direction of the maximum-speed daily wind gust(s) (in tens of degrees relative to north (0°)), variability (SD) of wind direction, mean daily wind gust speed (in km h$^{-1}$), variability (SD) of wind gust speed, and maximum daily wind gust speed, were calculated per month based on daily data from Environment Canada (*Government of Canada, 2018*) as tracked at the weather station in Parrsboro, NS. Water temperature data in the form of monthly mean, SD, minimum, and maximum sea-surface temperature (SST) values measured by satellite were obtained for the coordinates of Clarke Head from two different online databases: for 2008–2012, data were obtained from the online SST database of *Fisheries and Oceans Canada (2018)* (based on *Casey et al., 2010*), while for 2013–2017 they were obtained from that of the *International Research Institute for Climate and Society (2018)* (based on *Reynolds et al., 2002*). Air temperature provides a fair proxy for temperatures experience by crabs, especially at low tide, because for half of each day intertidal crabs are exposed to air rather than water temperatures (*Boudreau & Hamilton, 2012*; *Quinn, 2016*); although the exact temperatures experienced by crabs sheltering under macroalga-covered rocks at low tide are likely less extreme than actual air temperatures (e.g. *Scrosati & Ellrich, 2018*), these can still provide an index of temporal fluctuations in temperatures experienced by crabs at low tide. While not all crab life stages will be directly affected by SSTs (benthic juveniles and adults), others will (larvae), plus SST data are more readily available and well-validated than bottom temperatures are (*Reynolds et al., 2002*; *Casey et al., 2010*), and in a shallow, well-mixed area with strong tides like the Minas Basin SSTs should provide a reliable index of temporal fluctuations in temperature throughout the water column (*Hetzel et al., 2015*). Values of the NAO index for each month were obtained from the *Hurrell et al. (2003)* database (*National Center for Atmospheric Research Staff, 2018*). Oceanic circulation effects were assessed by extracting potential total annual settlement (= total summer

settlement) in the Minas Basin predicted by a bio-physical oceanographic model of larval dispersal, originally developed for American lobster (*Quinn, Chassé & Rochette, 2017*) but with development times of larvae modified for eastern North American green crabs (after *Dawirs, 1985*). Both potential settlement in the previous and same year as density sampling were considered as potential predictors of crab abundance.

Correlations of a grand total of 257 potential environmental predictor variables against intertidal crab densities in the years 2008–2017 were thus tested by calculating Pearson's correlation coefficient ($r$) values. Whether each correlation coefficient was significantly ($p \leq 0.05$) different from a value of zero (with $N = 68$ pairs) was first assessed (*Cohen, West & Aiken, 2003*; https://www.danielsoper.com/statcalc/). Any potential predictors that were not significantly correlated with intertidal crab density were not considered further; 159 predictors were eliminated at this step (see Supplemental Information). Ten additional potential biological predictor variables (population parameters such as sizes, sex ratios, etc.; Table 2) were also considered, but were excluded from these analyses because they were all strongly and significantly correlated with each other and most environmental predictors (see Supplemental Information), meaning that they would cause problems due to collinearity if included in multiple regression analyses (*Grafen & Hails, 2002*; *Anderson, 2008*); these variables were also significantly correlated with intertidal density (see 'Results'). The remaining 98 environmental predictors that were significantly correlated with intertidal density were then further examined. The predictor variables that were most strongly correlated with intertidal densities overall (i.e., those with the ten largest $r$-values), as well as the most-strongly correlated variable of each type of data (air temperature, SST, precipitation, wind speed, wind direction, NAO index, and potential settlement), were determined and examined in more detail; five predictors occurred in both of these groups.

An additional comparison of interest was whether interannual changes in model-predicted annual settlement agreed with observed abundances of juvenile crabs, as this might signal whether processes affecting larvae in the plankton or affecting newly-settled juveniles had a greater impact on early benthic recruitment at this site; therefore, the correlation between these values and its significance was also tested.

Next, a series of multivariate regressions were carried out to assess whether and to what extent multiple environmental factors may have impacted crab densities together. Because there were 98 potential predictor variables (most of which were significantly correlated with one another; see Supplemental Information) but the maximum number of predictors that could be tested on this dataset ($N = 68$ sampling trips) was nine, it was necessary to limit the analysis to simpler linear regression models and select a subset of potential predictors to be tested in them. Because many of the predictors among those with the top ten strongest correlations with intertidal density were similar types of variables (e.g., SST variables for the late winter-early spring months; see 'Results' and Supplemental Information), the set of seven predictors consisting of those of each type that were the most strongly correlated with density were used as the initial set of predictors in a series of stepwise multiple linear regression analyses (*Grafen & Hails, 2002*). An initial regression model was thus tested

that included all of these seven potential predictors as independent variables and intertidal density as the dependent variable (Table 3). Then, the independent variable with the highest, non-significant *p*-value was removed to generate a second model. This procedure was repeated until a model was produced that contained only significant independent variables. A null model (intercept-only) was also included so that other models could be compared against it. These statistical analyses were carried out in R v.3.1.1 (*Team R Core, 2014*).

All models were evaluated using various metrics to assess how well they and the environmental variables included within them were able to capture interannual changes in green crab intertidal densities at Clarke Head. Values of $r^2$ were generated to assess goodness of fit for each model, and Akaike's Information Criterion corrected for finite sample size ($AIC_C$) was calculated to determine the best model. $AIC_C$ is an index of the amount of useful information contained within a model, and the best model out of all those tested is the one with the lowest $AIC_C$ value (*Anderson, 2008*). $AIC_C$ is distinct from goodness of fit because though its value is decreased based on model fit, it is increased based on model complexity (e.g., number of parameters, $k$), thus penalizing models for potential biases due to overfitting and collinearity (*Anderson, 2008*). As correlations among predictor variables cause issues due to collinearity (*Grafen & Hails, 2002*; *Anderson, 2008*), whether the predictors included in the 'best' model were significantly correlated was also checked before concluding it was indeed the best model.

Model $\Delta_i$ and weights ($w_i$) were also calculated. $\Delta_i$ of a given model is simply that model's $AIC_C$ value minus that of the best model (so best model $\Delta_i = 0$) and gives a measure of information lost by not using the best model; in general if a model's $\Delta_i$ is greater than 12–16 it is extremely unlikely to contain useful information, but a model with $\Delta_i < 2$ may still contain useful information, even if it is not the 'best' model (*Anderson, 2008*). Model $w_i$ is the 'likelihood' of a given model (model likelihood = EXP($-0.5*$ $\Delta_i$)) divided by the sum of all tested models' likelihoods, and gives an indication of the 'distribution of evidence' among all models tested; if the best model has $w_i$ close to 1 there is little evidence that other models are useful, but if one or more non-best models has nonzero and relatively high weight then they may also be useful (*Anderson, 2008*). Finally, evidence ratios ($w_i$ of one model divided by that of another) were calculated between specific models of interest, particularly the null model and the best model, to assess relatively how much evidence there was in favour of one model over another (*Anderson, 2008*).

## Additional information from Local Ecological Knowledge

Local recreational fishers asked to provide data to calculate subtidal crab CPUE were also asked to comment upon any other occurrences of interest in the area over the course of the study period, and particularly in 2013. All provided similar anecdotal observations regarding changes in abundances and presence/absence of particular species in the area that are potential predators or prey of green crabs. These observations are briefly reported in the Results and discussed.

**Table 3  Results of stepwise multiple regressions done to find the best combination of environmental variables to explain interannual changes in intertidal green crab densities at Clarke Head, NS, from 2008–2017.** One of each type of environmental predictors (air temperature (AirTemp), sea-surface temperature (SST), precipitation (Precip), wind direction (WindDir), wind speed (WindSpeed), North Atlantic Oscillation index (NAOI), and potential settlement (AnnSettlement)) that had the strongest, significant correlation with intertidal density was used in these analyses (see Figs. 6A–6L). The average (Mean), standard deviation (SD), minimum (Min), and maximum (Max) daily values of predictor variables in different months (Jan–Dec) of the previous year (PY) and same year (SY) as densities were sampled were tested. The variables in each model in bold text were significant in the regression models used ($p \leq 0.05$). Progressively simpler models were tested, going from a 'full' model containing all seven potential predictors (model #1) to models (#2–6) with fewer variables, removing the non-significant predictor with the highest $p$-value at each step until only significant predictors remained (#6). An intercept-only ('null') model (#0) was also tested for comparison with other models. Also calculated to assess fit and select the best model were $r^2$ (fit), $AIC_C$ values (fit penalized based on the number of parameters, $k$, in a given model), $\Delta_i$, and AIC weight ($w_i$) values (see 'Methods' text and *Anderson, 2008*, for details); models were ranked based on $AIC_C$ values.

| # | Model: | $r^2$ | $k$ | $AIC_C$ | Rank | $\Delta_i$ | $w_i$ |
|---|--------|-------|-----|---------|------|-----------|-------|
| 0 | Density = **2.541** | 0 | 2 | 372.359 | 7 | 26.898 | $5.116*10^{-7}$ |
| 1 | Density = 9.433 − 0.105*MarSYMaxAirTemp + 1.856*JulPYNAOI + 0.127*JulPYSDPrecip + 0.041*AnnSYSettlement + 1.710*MarSYMaxSST − 0.858*OctPYMeanWindDir + 0.162*OctPYMeanWindSpeed | 0.407 | 9 | 353.780 | 6 | 8.319 | 0.006 |
| 2 | Density = 10.819 + 1.895*JulPYNAOI + 0.156*JulPYSDPrecip + 0.041*AnnSYSettlement + 1.571*MarSYMaxSST − 0.893*OctPYMeanWindDir + 0.118*OctPYMeanWindSpeed | 0.406 | 8 | 351.218 | 5 | 5.757 | 0.020 |
| 3 | Density = 17.726 + 1.916*JulPYNAOI + 0.184*JulPYSDPrecip + 0.045*AnnSYSettlement + **1.783*MarSYMaxSST** − 1.05*OctPYMeanWindDir | 0.405 | 7 | 348.722 | 4 | 3.261 | 0.069 |
| 4 | Density = 19.747 + 2.525*JulPYNAOI + **0.057*AnnSYSettlement** + **1.910*MarSYMaxSST** − 1.143*OctPYMeanWindDir | 0.403 | 6 | 346.458 | 3 | 0.997 | 0.215 |

*(continued on next page)*

**Table 3** (*continued*)

| # | Model: | $r^2$ | $k$ | AIC$_C$ | Rank | $\Delta_i$ | $w_i$ |
|---|--------|-------|-----|---------|------|-----------|-------|
| 5 | Density = 9.271 + **0.038\*AnnSYSettlement** + **1.272\*MarSYMaxSST** − 0.570\*OctPYMeanWindDir | 0.390 | 5 | 345.575 | 2 | 0.114 | 0.335 |
| 6 | Density = −1.312 + **0.025\*AnnSYSettlement** + **0.831\*MarSYMaxSST** | 0.369 | | 345.461 | 1 | 0 | 0.355 |

# RESULTS

## Interannual changes in overall crab densities

The density of green crabs at Clarke Head, NS observed on the intertidal zone differed significantly overall among years from 2008–2017 (ANOVA, $F_{9,58} = 4.461$, $p < 0.001$; Fig. 2). Intertidal densities increased from 2008 to 2010, when peak abundances were observed, remained relatively high in 2011 and 2012, and then rapidly declined to all-time lows in 2013 (Fig. 2). Densities remained low after 2013, but did gradually increase from 2014 to 2017 (Fig. 2). Densities were also highly variable among samples in 2008–2012, but from 2013 onward they were consistently low (compare error bars in Fig. 2). Intertidal densities were highest and above-average in the years 2010–2012, and lowest and below-average from the years 2013–2016 (Fig. 2). Intertidal densities in 2010 were significantly higher than those in 2013, 2014, and 2015 (Tukey's HSD test, $p \leq 0.034$) (Fig. 2). The difference between densities in 2010 and 2012 was marginally non-significant (Tukey's HSD test, $p = 0.053$), and differences among densities in all other years were not statistically significant (Tukey's HSD test, $p \geq 0.096$) (Fig. 2). Average intertidal densities in 2013 in particular were much lower (by 13.3–91.4%) than those in other years; 2013 densities were significantly lower than those in 2010 and 2012 (Dunnet's test, both $p = 0.001$), but did not significantly differ from those in any other year before (2008, 2009, 2011: Dunnet's test, $p \geq 0.172$) or after this (2014–2017: Dunnet's test, $p \geq 0.809$) year (Fig. 2).

## Changes in population size structure, sex ratios, and recruitment among years

Green crabs of nearly all possible sizes, from juveniles $\leq 5$ mm CW up to adult males of 86 mm CW, were found at Clarke Head, NS over the ten years of sampling, although crabs larger than 50 mm CW were always uncommon on the intertidal (Figs. 3A–3J). In most years preceding 2013, 83.4–91.0% of the crabs observed were mature (i.e., $\geq 1$–2 years old and with CW $\geq 15$ mm) (Fig. 4A), a wide range of crab sizes were observed (Figs. 3A–3E), and overall average sizes of crabs were relatively typical or large for an intertidal population (28.2–52.8 mm CW; *Boudreau & Hamilton, 2012*) (Fig. 4B). Notably higher proportions of sampled crabs in larger size classes were observed in 2010–2012 than in other years (Figs. 3A–3E), the same years in which crab abundances were also highest (Fig. 2). In 2013, a dramatic shift occurred in which the proportion of mature crabs in the area plummeted to 58.1% (Fig. 4A), mean sizes of all crabs found decreased to 15.7 mm CW (Fig. 4B), and crabs of CW > 30 mm almost completely disappeared from the intertidal (Fig. 3F). In

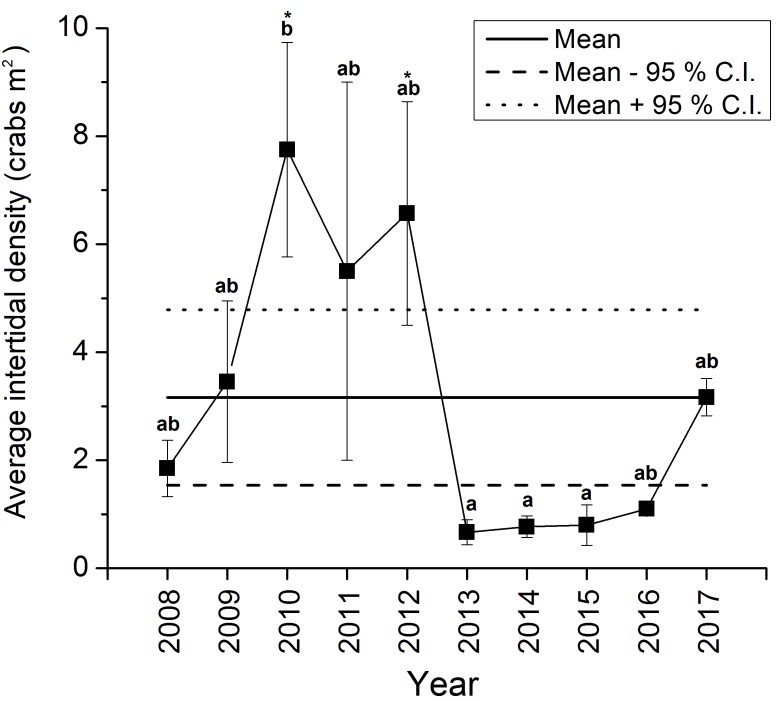

**Figure 2** **Average ± standard error (SE) annual intertidal densities (individuals m$^{-2}$) of green crabs in the Clarke Head population in different years from 2008 to 2017.** The solid horizontal and dashed lines are the overall mean ± 95% confidence intervals (C.I.s) of intertidal density from 2008–2017. Different letters above yearly means indicate years with significantly different crab abundances (Tukey's HSD test, $p \leq 0.05$). Asterisks ('*') indicate years in which crab abundance was significantly different (i.e., greater) than that observed in 2013 (Dunnet's pairwise multiple comparison $t$-test, $p \leq 0.05$).

subsequent years (2014–2017) larger crabs (up to 77 mm CW) began to reappear, but still remained much less abundant than in previous years (Figs. 3G–3J), and the proportion of mature crabs (56.3–68.8%) and mean overall crab sizes (17.1–26.6 mm CW) remained lower than they were before 2013 (Figs. 4A, 4B). Further, even with this gradual return of larger crabs total crab abundances (densities) remained low in 2014–2016 (Fig. 2); although in 2017 intertidal density did climb to a value near the overall average (Fig. 2). Mean sizes and size distributions of male, female, and ovigerous female crabs similarly shifted toward smaller sizes in 2013–2016 than in 2008–2012, with some sign of returns to pre-2013 values occurring in 2017 (Figs. 3F–3J, 4A, 4B).

The intertidal population sampled was in general female-dominated (average proportion of mature crabs female ± SE = 0.580 ± 0.082, maximum = 0.700 in 2010), except in one year (2014, proportion females = 0.438) (Fig. 4A). Sex ratios in this population fluctuated considerably among years, but with no clear overall pattern except perhaps a slight decrease in the proportion of females among sampled crabs from 0.670–0.700 in the earliest years sampled (2008–2010) to 0.438–0.614 in later years (Fig. 4A). About 50% of females observed were ovigerous in all years (average proportion = 0.540 ± 0.083, range = 0.357–0.629), and this proportion fluctuated among years without any clear pattern except that ovigerous females were notably rarer in 2012 than in other years (Fig. 4A). Ovigerous

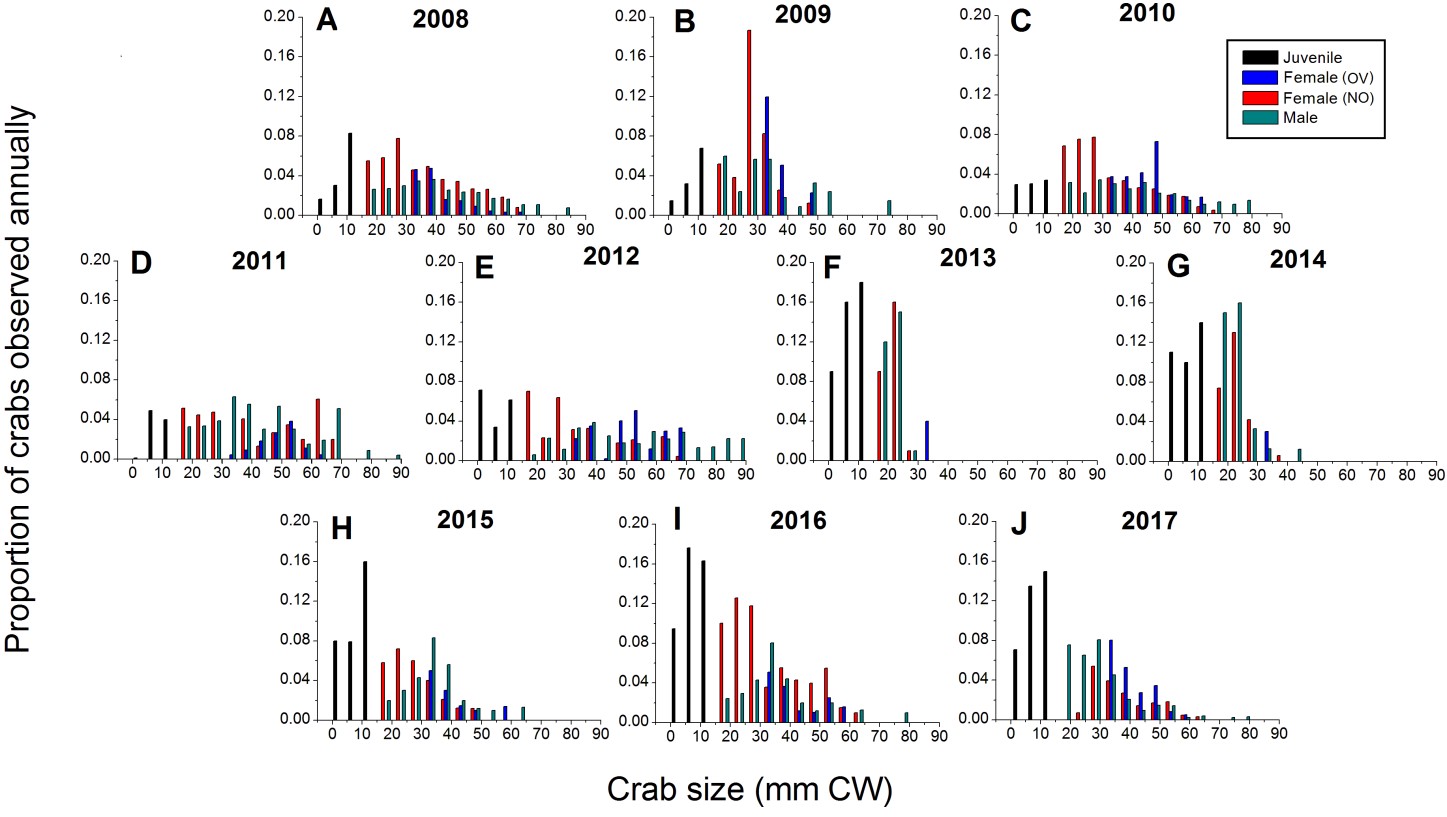

**Figure 3** **Detailed size structure of green crabs observed on the intertidal at Clarke Head, NS, in each year monitored (2008–2017).** Each plot shows the proportion ($y$-axes) of all crabs observed in each year from (A) 2008 to (J) 2017 (summed across all sampling trips) that fell within each five mm carapace width (CW) size bin ($x$-axes). Differently coloured bars indicate juvenile (black), ovigerous female (OV, blue), non-ovigerous female (NO, red), and male (green) crabs. All bars in each yearly plot should sum to 1.0.

females tended to be larger than non-ovigerous ones in this population, and also were often larger on-average than males (Figs. 3A–3J, 4B), though it should be noted that very large males were also often present (Figs. 3A–3J).

The abundance of juvenile (CW < 15 mm) crabs observed on the intertidal at Clarke Head also varied considerably among years, but with no clear pattern (Fig. 5A). Juvenile numbers were notably very low in 2011 and 2013, and very high in 2012 (Fig. 5A). Subtidal CPUE of large crabs was relatively high and above-average from 2008–2012 and relatively low and below-average from 2013–2017 (Fig. 5B). Subtidal CPUE thus underwent similar fluctuations to intertidal densities (Fig. 2), although the year 2008 was an exception in which intertidal densities were relatively low (Fig. 2), while subtidal CPUE was average (Fig. 5B).

**Correlations among biological characteristics and intertidal densities**
All biological characters of the sampled crab population discussed above (sizes, sex ratios, etc.) were strongly and significantly correlated with observed intertidal crab densities (Table 2). Specifically, mean sizes of all, male, all female, ovigerous female, and non-ovigerous female crabs, the proportion of the crab population mature, the proportion of

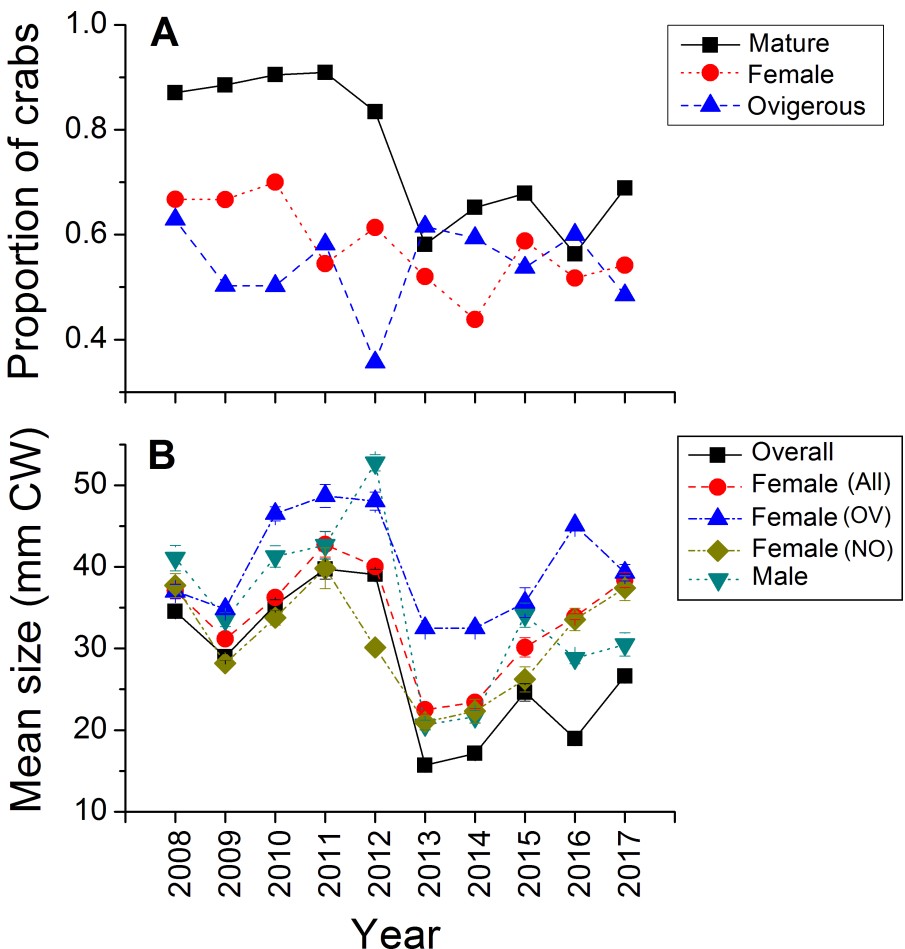

**Figure 4 Demographic characteristics (proportions in different reproductive categories or sizes) of the green crab population at Clarke Head, NS, in each monitored year.** Plots show: (A) the proportion of all crabs observed in each year (summed across all sampling trips) that were mature (CW ≥ 15 mm; black squares), the proportion of all mature crabs that were female (red circles), and the proportion of all female crabs that were ovigerous (blue triangles); (B) the mean ± SE size (mm CW) of all crabs in each year overall or fit into different categories (represented by different symbol types).

mature crabs that were female, and juvenile abundances were positively correlated with intertidal density (Table 2). Interestingly, however, the proportion of female crabs that were ovigerous was strongly negatively correlated with intertidal density (Table 2), which was likely driven in large part by the relatively low abundance of ovigerous females in 2012 (Fig. 4A). Intertidal densities and subtidal CPUE showed similar fluctuations over time, and thus were positively correlated with each other between 2008 and 2017 ($r = +0.521$, $N = 68$, $p < 0.001$). Observed juvenile abundance and model-predicted annual settlement were likewise positively correlated ($r = +0.375$, $N = 10$, $p = 0.286$), although since the correlation between these two metrics was relatively weak and non-significant this likely indicates some decoupling between pre- and post-settlement processes affecting new crab recruit numbers in nature (see Fig. 5A).

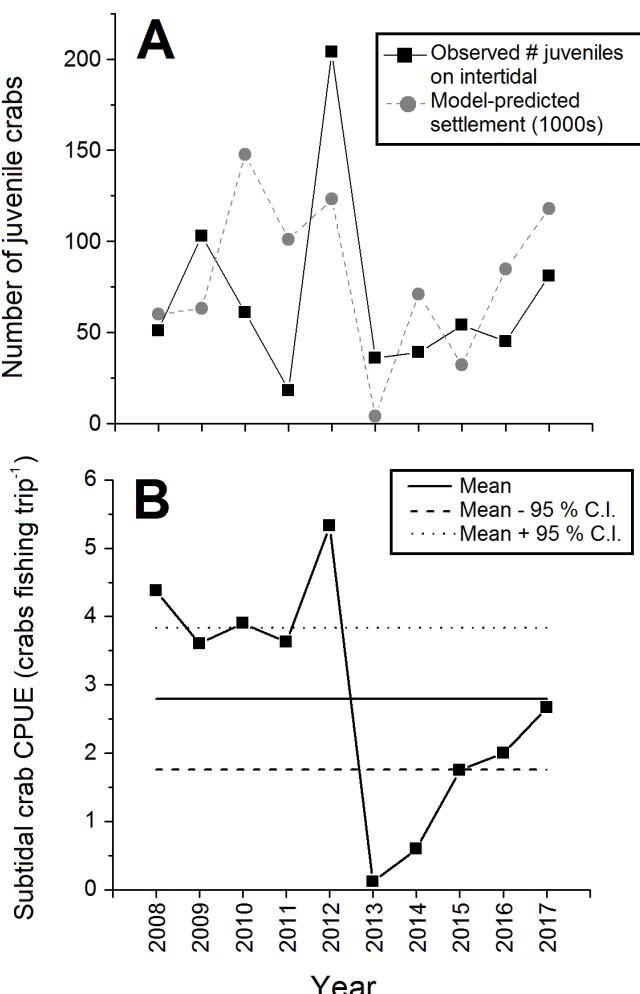

**Figure 5** **Abundance of juvenile and large subtidal crabs in the monitored green crab population at Clarke Head in different years from 2008–2017.** Plots show: (A) Total annual observed numbers of juvenile (CW ≤ 15 mm) crabs (black squares) and model-predicted numbers of settling crab larvae (1,000s of megalopae; gray circles); and (B) catch-per-unit-effort (CPUE) of large crabs in the subtidal zone (number of crabs caught as bycatch per fishing trip by recreational fishers). In (B) the solid and dashed horizontal lines represent the overall mean ±95 % confidence intervals (C.I.) of subtidal CPUE across all years.

## Potential correlates and causes of interannual changes to green crab abundances

Of the 257 environmental variables considered as potential predictors of changes in green crab abundances across years, 98 were significantly correlated (i.e., $r < -0.238$ or $r > +0.238$, $N = 68$, $p \leq 0.05$) with intertidal crab densities (Supplemental Information). All of the predictors among those with the top ten strongest correlations had positive and significant $r$-values $\geq 0.5$ (Figs. 6A–6I). The strongest correlation was a positive one between the maximum SST during March of the same year and intertidal density ($r = +0.560$, $N = 68$, $p < 0.001$; Fig. 6A). Five of the remaining predictors out of those with the ten strongest correlations with intertidal density included SST parameters for the

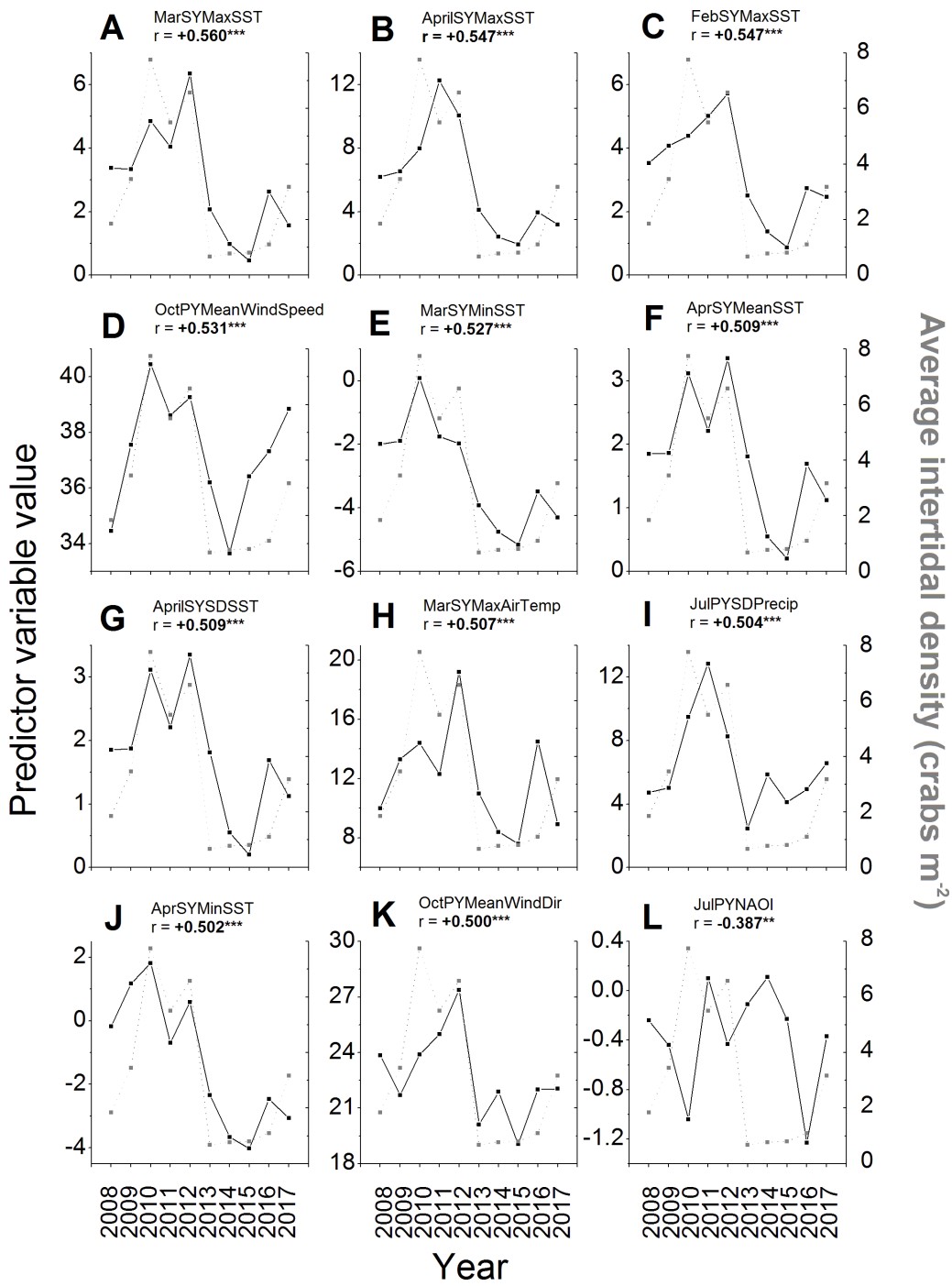

**Figure 6** **Environmental variables most strongly correlated to intertidal densities across years.** The values of the potential environmental predictor variables (left *y*-axes and black squares) examined in each year (*x*-axes) that resulted in the first (A) to 12th (K) strongest correlations with average annual intertidal crab densities (right *y*-axes, gray squares) of those tested (see Supplemental Information for all correlations). The sixth strongest correlation of all was found for total predicted (continued on next page...)

**Figure 6 (…continued)**
settlement in each year (see Fig. 5A and 'Results' text). The NAO index variable with the strongest ob-
served correlation to intertidal density (36th highest $r$-value) is also shown (L). For details and sources of
data, see the 'Methods' text. The average (Mean), standard deviation (SD), minimum (Min), and maxi-
mum (Max) daily values of predictor variables in different months (Jan–Dec) of the previous year (PY)
and same year (SY) as densities were sampled were tested. These included air temperature (AirTemp), sea-
surface temperature (SST), precipitation (Precip), wind direction (WindDir), wind speed (WindSpeed),
North Atlantic Oscillation index (NAOI), and potential settlement (AnnSettlement). Significant correla-
tions ($r < -0.630$ or $r > +0.630$, $N = 10$, $p \leq 0.05$; *Cohen, West & Aiken, 2003*; https://www.danielsoper.
com/statcalc/) are indicated by bold text and asterisks as follows: *, $0.01 < p \leq 0.05$; **, $0.001 < p \leq 0.01$;
***, $p < 0.001$.

early months (February–April) of the same year as that in which densities were measured
(Figs. 6B, 6C, 6E–6G; Supplemental Information), specifically the maximum February
SST, minimum March SST, and the mean, SD, and maximum of April SST. All of these
SST predictors were positively correlated with density, meaning that higher SSTs during
late winter-early spring led to higher densities, while lower SSTs led to lower densities
(Figs. 6B, 6C, 6E–6G). The top ten predictors most-strongly correlated with density also
included the mean wind speed in the previous year's October (Fig. 6D), total annual
settlement in the same year ($r = +0.520$, $N = 68$. $p < 0.001$; compare Fig. 2 vs. Fig. 5A),
maximum air temperature in March of the same year (Fig. 6H), and the variability (SD)
of total precipitation during July of the previous year (Fig. 6I), all of which were positively
correlated with density. The wind direction variable most strongly correlated with density
was the average wind direction in October of the previous year, which had the 12th highest
$r$-value overall and was positively and significantly correlated with density (Fig. 6K). The
most-strongly correlated NAO index variable was that for July of the previous year, which
had the 36th highest $r$-value overall (see Supplemental Information) and was significantly
and negatively correlated with density (Fig. 6L).

Additional interesting predictor variables showed correlations with lower, but still
significant values (i.e., $r$ between $\pm 0.238$–0.5). For example, most SST and air temperature
variables for the winter, spring, and summer months of the same year were positively
correlated with intertidal density (see Supplemental Information), meaning that colder
winters, springs, and summers resulted in lower crab numbers; however, several mean and
minimum SSTs and air temperatures during the summer and early fall of the previous year
(July–October) were significantly negatively correlated with density (see Supplemental
Information), meaning that warmer summers resulted in lower intertidal densities in
the following year. SSTs were generally better predictors of intertidal density than air
temperatures were, while precipitation and especially wind variables were overall less well-
correlated with densities than these (Figs. 6A–6C, 6E–6H, 6J; Supplemental Information).
Crab density in a given year was also significantly correlated to the value of the North
Atlantic Oscillation (NAO) index in certain months within the summer or early autumn of
the same or previous year, but not in other seasons (Fig. 6L; see Supplemental Information).
Specifically, higher or more positive values of the NAO index in the preceding year's July
and August and the same year's June and August were associated with lower crab densities
in a given year, while higher NAO index values in the preceding year's September were
associated with higher crab densities (see Supplemental Information); all other periods' NAO index values were not significantly correlated with intertidal densities.

Stepwise multiple regression resulted in six models to describe changes in intertidal crab densities across years based on up to seven potential predictors, which represented the best individual predictors (i.e., highest $r$-values) of each type of variable examined (Table 3; Supplemental Information). The best model (#6) explained 40.7% of the interannual variation in intertidal density by relating it to two environmental predictor variables: total annual potential settlement and maximum March SST, both in the same year (Table 3). Both predictors included in the best model were not significantly correlated with one another, although some of those included in other non-best models were (see Supplemental Information). The variables in the best model therefore provided perhaps the best explanation of observed changes to intertidal crab densities. However, all models were much better than a null (#0, intercept-only) model (evidence ratio of all models versus the null = 10,823.771–693,128.818) and had comparable $r^2$-values (0.369–0.407) to each other (Table 3). Also, although the best model (#6) carried most of the model weight ($\Delta_i = 0$, $w_i = 0.355$), the second- and third-best models may also have contained additional useful information (Table 3). Specifically, the second-best model (#5), containing the same predictors as the best model in addition to the mean wind direction in October of the previous year, had an $AIC_C$ value almost the same as that of the best model ($\Delta_i = 0.114$) and a model weight of 0.335 (Table 3). The third-best model (#4), containing the same predictors as model #4 plus the NAO index value in July of the previous year, also had a $\Delta_i$-value <2 ($\Delta_i = 0.997$) and carried some weight ($w_i = 0.215$; Table 3). Therefore these other variables may also have had some influence on interannual fluctuations in this crab population.

## Local Ecological Knowledge of changes in the study area

In addition to green crab bycatch during fishing being high prior to 2013 and then very low afterwards, recreational fishers reported changes to other bycatch finfish species that may have implications to green crabs. Longhorn sculpin (*Myoxocephalus octodecemspinosus*) were always a frequent bycatch species in this area, but from 2013 onward they became an extremely abundant nuisance species to fishers. Cunners (*Tautogolabrus adspersus*) were first caught by fishers off of Clarke Head in 2012 and have been present in the area since, but before this no fisher interviewed had ever encountered one. Many of the fishers consulted also frequently harvested marine worms from sand- and mudflats in the area, a practice often done at nighttime low tides. A frequent observation by local marine worm harvesters in the area prior to 2013 was that during nighttime low tides green crabs were extremely abundant on sand- and mudflats, where they frequently preyed on marine worms (e.g., *Alitta virens*). However, from 2013 onwards encounters with crabs on such excursions become very rare. Further, the marine worms used as bait by fishers ('shad-worms', a local name around Parrsboro, NS for mainly sandworms such as the king ragworm, *Alitta virens*; Wilson Jr & Ruff, 1988; Miller, 2009) were also observed by local recreational fishers to become extremely rare in 2013 in sand- and mudflat areas near Clarke Head in which they were previously abundant, and have remained so since then. Locals were of the opinion

that excessive predation by green crabs led to the collapse of worm stocks in 2013, which in turn led to crab starvation and die-off.

## DISCUSSION

The present study presents a ten-year dataset obtained by monitoring a single, established population of invasive green crabs (*Carcinus maenas*) in the Minas Basin of the upper Bay of Fundy, Atlantic Canada. Monitoring of densities, size structure, sex ratios, and reproductive ratios of intertidal crabs revealed a marked wholesale decline in this population occurring within the year 2013. This decline in 2013 was associated with a shift in the population's demographic structure towards smaller-sized and fewer sexually-mature or reproductive (i.e., ovigerous) crabs, and these changes have largely persisted through subsequent years. These changes to the crab population were significantly correlated with various environmental variables, suggesting some change to the physical conditions experienced by crabs in the area during or after 2013. Although correlations do not equal causation, the links established herein between interannual changes in crab biology and their physical environment are highly suggestive, and provide directions that can be pursued by future research.

There are several important aspects of the monitoring program that could be improved in the future and may have led to some uncertainty regarding specific numbers (e.g., crab densities, water temperatures) in the present study. Most importantly, the use of daytime intertidal quadrats to simplify the monitoring of the population has the potential to have missed important trends and to be inconsistent with results of other studies of crabs in the study region that used more standard methodologies (e.g., Fukui or pitfall traps; *MacDonald et al., 2018*). Crabs would almost certainly have been more abundant and present over other areas of the intertidal (not just in shelters among reefs) if sampling had been done at night, when crabs are more active and likely to stay on the intertidal due to lower air temperatures, water stress, and visibility to terrestrial predators compared to daytime (*Crothers, 1968*; *Klassen & Locke, 2007*). LEK observations reported by local fishers of reduced crab activities during nighttime worm harvesting activities in and after 2013 do suggest that, even if sampling had been done at night, the reduction in crab numbers in 2013 would have been observed; although this is unconfirmed at present. The use of simplified intertidal quadrats instead of standard Fukui traps in this study means that the densities quantified herein are not directly comparable to those collected in other studies in the region (e.g., *MacDonald et al., 2018*), which does impair the possibility of direct comparisons across studies. Crabs also undergo tidal migrations (*Hunter & Naylor, 1993*), and because sampling focused mainly on intertidal crabs no information was obtained regarding what proportion of the crab population left the intertidal zone for the subtidal on ebb tides, and whether this changed among years. Declines in the CPUE of large (>50 mm CW) subtidal crabs caught during flounder fishing trips in and after 2013 were noted, which lent support to the conclusion that this decline was a real event that impacted the entire local crab population, and not just the result of crabs migrating away from the intertidal for some reason more in later years (*Hunter & Naylor, 1993*). However, because the catchability

of green crabs by flounder fishing hooks was not formally quantified (though it did seem to be high, see Methods) and the proportion of crabs leaving the intertidal at low tides was unknown, these results are not without some degree of uncertainty. Future monitoring work within this population should attempt to quantify these variables, as well as attempt to sample crab densities at night and using Fukui traps, to confirm the conclusions drawn herein and improve the quality of future data obtained. A more thorough and formal analysis of spatial and temporal variations in crab numbers at this site (among tidal elevations (*Scrosati, Grant & Brewster, 2012*), sampling dates within a year, etc.) would likely also be worthwhile. Direct monitoring of environmental variables at the site, such as temperatures (air at low tide only, and water at high tide), could also improve future analyses of the drivers of population changes. However, while specific numbers may not have been assessed with perfect accuracy, the magnitude of the change observed at and after 2013 as captured over this ten-year monitoring program is unequivocally a real and important result.

The lack of substantial recovery of this population to its former characteristics from 2014 to 2017 could be a result either of slow growth due to reduced reproductive capacity (i.e., loss of larger, mature individuals), and/or due to the cause of the initial decline still being in play within the environment inhabited by these crabs. Better understanding of the potential causes of this decline is needed to fully examine the scope and implications of this change. However, the negative implications of the observed loss of breeding stock to this population's future growth and stability is obvious, particularly with the loss of larger individuals that are thought to have considerably greater size-specific-fecundity in decapod crustaceans than smaller breeders (*Somers, 1991*; *McGaw, Edgell & Kaiser, 2011*). The ability of crabs to feed on shelled prey is also strongly size-dependent, such that smaller crabs have access to a more limited suite of prey than larger one (*Elner & Hughes, 1978*), so the shift in population size structure may also have implications to crab foraging ecology and to prey species and their communities in the region. Even in 2017, when observed overall crab densities appear to have begun approaching 'normal' levels for this study site and period and some large crabs were present on the intertidal, juvenile crabs still made up a substantial portion of the population and average sizes were smaller than in pre-2013 conditions. Whether this population will fully recover and how long it will require to do so are open questions, but because crabs interact strongly with many species (e.g., *Klassen & Locke, 2007*; *Boudreau & Hamilton, 2012*) if recovery is slow or does not occur the ecology of the study area could change considerably; thus monitoring should and will continue.

It is important to note that intertidal crab densities at Clarke Head were comparably low in 2008 to those seen in and after 2013, and these low densities were associated with some similar environmental factors to those putatively involved in the 2013 decline (e.g., oceanographic currents leading to low predicted settlement). However, lower abundances in 2008 were not associated with differences in the size or reproductive structure of this population or low subtidal CPUE versus other years, whereas in 2013 larger, sexually mature crabs become extremely scarce and subtidal CPUE was very low. One could postulate some similar factors having been in play in 2007–2008 to those in 2012–2013, but with additional factors from 2012–2013 and thereafter (discussed below) leading to particularly high losses

of larger, breeding crabs. As a result, the population recovered quickly in 2009 from 2008 declines, but as of 2017 had not fully-recovered from the 2013 collapse.

Results of this study showing population declines of a benthic marine invertebrate in 2013 agree with findings of other studies and species in the Bay of Fundy-Gulf of Maine region. *Kienzle (2015)* and *MacDonald et al. (2018)* noted a decline in abundances of green crabs inhabiting sites in Chignecto Bay, a nearby but distinct branch of the upper Bay of Fundy to the Minas Basin (Fig. 1), from 2013 to 2014. Monitoring of early benthic recruitment (i.e., settlement) of American lobster (*Homarus americanus*) juveniles by the American Lobster Settlement Index (ALSI) recorded record-low recruitment of juveniles in the year 2013 throughout most of the Bay of Fundy, Gulf of Maine, and southwestern Nova Scotia (*Wahle & Carloni, 2017*). Lobster recruitment since 2013 in these areas has remained low compared to its pre-2013 values and appears to have been even lower than in 2013 in the year 2016 (*Wahle & Carloni, 2017*). Potential settlement of lobster larvae in the Gulf of Maine-Bay of Fundy system and the Scotian Shelf has also been predicted by a bio-physical dispersal model of the species' range (*Quinn, Chassé & Rochette, 2017*) to have been extremely low in 2013 (BK Quinn, J Chassé, and R Rochette, pers. comm., 2017). Benthic recruitment of soft-shell clam (*Mya arenaria*) settlers monitored on mudflats in the Bay of Fundy was also extremely low in 2013 compared to earlier years (*Clements, 2016*). Along with the present study, these various observations point to the occurrence of some sort of large-scale oceanographic event in the region of the Bay of Fundy during or around the year 2013. This event appears to have negatively impacted numerous marine species in the region, and as of 2017 many species' populations have still not completely recovered. If such an event occurred, what were the causes or mechanisms? Results of the present study and postulated causes of declines reported in others may provide some hints, as discussed below.

Recent climate change has had important impacts on thermal variability within the atmosphere and the ocean, including warmer summers and periodically colder, harsher winters in temperate regions such as Atlantic Canada (*Petrie, 2007*; *Mills et al., 2013*; *Kienzle, 2015*; *MacDonald et al., 2018*). When extremes of seasonal temperatures coincide due to such climatic changes, the effects on biota can be considerable (e.g. *Quinn, 2016*). Excessively hot summers in recent years may have played a role in observed declines because heat stress leads to decreased performance and survival in decapod crustaceans, especially in larger-bodied benthic life stages that tend to have lower heat tolerances (e.g., *Byrne, 2011*). The Gulf of Maine is now widely acknowledged to have experienced an 'ocean heat wave' in 2012, which led to negative heat stress effects on many species, and changes to phenology (e.g., moulting by lobsters) of many others (*Mills et al., 2013*). This heat wave likely contributed to results of the present study, as higher temperatures during the summer months of a given year were correlated with lower crab densities in the following year, with 2012, the year preceding 2013, having one of the hottest summers recorded (see Supplemental Information). Of course, crab densities in the monitored population were actually quite high during the summer of 2012, implying that rather than being stressed crabs were experiencing temperatures close to optimal for their survival and growth. Although it is important to note that ovigerous females were less frequently encountered

on the intertidal in 2012 compared to other years, which may mean that this very warm summer had negative impacts on the survival or reproduction of females. Negative effects of the 2012 heat wave on crab abundances in 2013 may alternatively have occurred via some indirect means, such as effects on prey or predator species, altered phenology, or perhaps accumulated sub-lethal stress effects from the previous year if supra-optimal temperatures had occurred.

Cold winters and springs may also have contributed to crab declines in 2013 onward, since crab survival at all life stages—but especially overwintering survival of young-of-the-year—is adversely affected by low (<6–10 °C) temperatures (*Berrill, 1982*; *Klassen & Locke, 2007*). Therefore, *Kienzle (2015)* and *MacDonald et al. (2018)* postulated that especially harsh, cold conditions during the winter from 2013–2014 likely contributed to the declines they observed in green crab numbers in Chignecto Bay. In the present study, crab intertidal densities in a given year were also found to be positively correlated with mean temperatures during the preceding winter and early spring months (i.e., colder winter and spring = fewer crabs); while the data examined herein do not suggest that the winter of 2013 was particularly cold (at least for the studied part of the Minas Basin), it was colder than that of preceding years, and the winters of 2014 and 2015 were notably cold (see Figs. 6A–6C, 6E–6H, 6J). However, it is possible that the combined negative effects of the very warm summer of 2012 on larger adult life stages and the somewhat cold winter of 2013 on earlier life stages could have led to some of the declines observed; this disparity in thermal extremes may also explain why numbers of larger crabs, which would have overwintered and experienced both extreme temperature conditions, were observed to have declined relatively more than smaller crabs in 2013.

Other environmental parameters found in this study to be related to lower crab abundances included greater wind speeds from a more southeasterly (i.e., towards a more northwestern direction, of higher degree values relative to north) in the preceding autumn (October) and greater variability in precipitation during the early summer (July) of the previous year (for more details, see 'Results', especially Table 3 and Figs. 6D, 6I, 6K). The exact roles of these factors and other wind- and precipitation-related predictors (see Supplemental Information) are complex, as different impacts in different seasons may affect one or more different life stages. Winds and precipitation, for example, may signal storm frequency, with more frequent storms being expected to have negative impacts on both planktonic larvae and benthic adults and juveniles (e.g., *Jury, Howell & Watson III, 1995*; *Moksnes et al., 2014*). Wind speed and direction impact surface currents and water circulation, so during the summer and autumn these can impact whether and how many larvae are retained versus dispersed (*Bertness, Gaines & Wahle, 1996*), thus affecting settlement and recruitment to benthic populations (*Crothers, 1967*; *Moksnes et al., 2014*). Reduced salinity resulting from high precipitation can adversely affect physiological health and survival of larvae in the surface waters directly and benthic life stages indirectly via runoff (*Nagaraj, 1993*; *McGaw, Reiber & Guadagnoli, 1999*; *Klassen & Locke, 2007*). The positive correlations found in this study of crab abundance with summer precipitation and autumn winds are somewhat unexpected. It is conceivable that these atmospheric forces alter circulation at or beneath the sea surface in such a way that is actually beneficial

to crabs—for example, by directing settling larvae shoreward or forcing saline water to flow into the estuary in response to wind-induced surface currents. Indeed, given the orientation of Clarke Head (Fig. 1C) more southeasterly winds in the autumn, blowing towards the northwest, would potentially help to direct larvae in surface currents toward the shore at this site and facilitate greater settlement there, while winds in the opposite direction could impede larval transport toward the coast. The strong positive correlation between model-predicted settlement—essentially a proxy for oceanography—and crab abundances lends some support to this supposition. However, this requires further study of the interaction of weather and oceanography in the study area to be confirmed and clarified.

It is interesting that in years in which oceanographic conditions leading to higher potential settlement occurred the abundance of crabs older than one year (>15–20 mm CW; *Crothers, 1967*; *Crothers, 1968*; *Klassen & Locke, 2007*) was also higher, and conversely when there was less settlement older crab abundances were also low. Clearly decreased settlement cannot explain low mature crab abundances in years such as 2013. However, these results do offer the intriguing suggestion that oceanographic conditions favouring settlement of larvae in the studied region also favour survival of older crabs. Given that the study site is located in a relatively shallow, well-mixed estuary, it is conceivable that this might be the case.

If indeed some large-scale oceanographic event was responsible for crab declines in 2013 and declines of other species in the region, one of the environmental variables examined which would be expected to have had an impact was the value of the NAO index. The winter NAO in particular has been found to be strongly associated with climate and ocean characteristics (*Hurrell et al., 2003*), such that in years with highly negative values of the winter NAO index bottom waters in the Bay of Fundy-Gulf of Maine system (including the southwest NS Shelf) have relatively low salinity and are colder than average, whereas in years with strong positive winter NAO index values waters in this region become warmer and more saline (*Petrie, 2007*). Negative winter NAO index-years also tend to have more frequent storms and stronger offshore-directed currents in this region than positive years (*Hurrell et al., 2003*; *Petrie, 2007*). The year 2013 had somewhat negative NAO index values during the winter months (see Supplemental Information), which may mean that conditions in the Bay of Fundy were unfavourable for survival of benthic marine species and retention and recruitment of their planktonic larvae; observations cited for various species in this region certainly suggest that NAO is negatively correlated to abundance and population recruitment (*Kienzle, 2015*; *Clements, 2016*; *Wahle & Carloni, 2017*). However, even greater negative values of the NAO index occurred in the winter of 2010, when summer crab densities observed herein were at their highest. Indeed, while significant correlations between crab densities and NAO index values for the summer months were found in this study, most were negative, such that more negative NAO indices were associated with higher crab abundances. However, NAO index values in winter months and other seasons were not significantly correlated with crab densities. This is unexpected, but does agree with other correlations (e.g., the positive relationship between precipitation and wind speeds during spring and summer months and crab densities; see Supplemental Information).
Clearly the relationship between the NAO index and localized recruitment is thus not as simple or direct in all cases as might be supposed. Perhaps the Minas Basin, being relatively isolated from the rest of the Bay of Fundy-Gulf of Maine system, is less affected by NAO-associated shifts in regional oceanography or climatology. Future studies should further examine the relationship between the NAO index and recruitment in other marine species in the region to clarify these complexities.

Crab populations could alternatively have declined due to biotic changes in the studied region, such as those cited by local recreational fishers concerned with other species potentially interacting with crabs. Increased presence in the area of two finfishes known to feed on benthic crustaceans such as smaller crabs (*Klein-MacPhee, 2002*; *Munroe, 2002*) in 2012–2013 may have played a part in crab declines. Cunner (*Tautogolabrus adspersus*) were largely absent from the inner Bay of Fundy prior to 2012 (*Munroe, 2002*), but then in 2012 they began to penetrate deeper into the bay (*Woodard, 2018*). Cunner numbers in the Bay of Fundy have since been lower than those in 2012, but nonzero (*Woodard, 2018*). Longhorn sculpin (*Myoxocephalus octodecemspinosus*) are native to the Minas Basin (*Klein-MacPhee, 2002*; *Parker, Westhead & Service, 2007*), but from 2013 onward they have become extremely abundant in the Minas Basin. The extent to which these fish species prey on green crabs and their impact on crab populations needs to be better quantified to determine whether predation by them could be responsible for crab declines. Declines to populations of species preyed on by green crabs in the area, such as the decline in sandworm (*Alitta virens*) stocks reported by recreational harvesters, may also be related to crab declines. Green crabs will prey on sandworms (*Crothers, 1968*; *Klassen & Locke, 2007*), so it is quite possible that predation by the invasive crab population on these worms in combination with worm harvest (potentially overharvest) by humans (*Miller, 2009*) could have contributed to the worm population's collapse (though this needs further study). However, green crabs are quite capable of feeding on a wide variety of prey species also present off of Clarke Head, including blue mussels (*Mytilus* spp.), barnacles (*Semibalanus balanoides*), periwinkles (*Littorina* spp.), dogwhelks (*Nucella lapillus*), soft-shell clams (*M. arenaria*), and many others (*Klassen & Locke, 2007*). All of these species were present and quite abundant throughout the 2008–2017 period (BK Quinn, pers. obs., 2013–2017), so even if crabs had relied heavily on sandworms as a food source they should have been readily able to switch their prey to any of the various other species present (*Huntingford & Taylor, 1997*) when the worm stock declined in 2013. While unconfirmed, these observations by locals provide interesting directions for future research to pursue.

Some additional or alternative localized factors that may also have been impacting crab populations within the Minas Basin during or since 2013 could include: development of tidal power within the Minas Channel, the only means of connectivity between the Minas Basin and the remainder of the Bay of Fundy (*Parker, Westhead & Service, 2007*; *Copping et al., 2016*); increasing eutrophication in recent years within the adjacent estuary, causing harmful algal blooms ('red tides') that cause anoxia and produce toxins potentially harmful to crustacean development (e.g., copepod: *Miralto et al., 1999*); effects of ocean acidification on crab exoskeleton formation and moulting (*Clements, 2016*; *Miller et al., 2016*); changes in the phenology (timing) of planktonic productivity on which larval diet

and survival depends (*Scrosati & Ellrich, 2016*; *Wahle & Carloni, 2017*); or others. However, the impact of all of these listed factors requires further directed study before a link to crab declines can be drawn.

## CONCLUSIONS

To conclude, there is much evidence that is suggestive that some sort of physical oceanographic and/or climatic event occurred in or around 2013 and impacted marine biota in the Bay of Fundy and Gulf of Maine, and this should certainly be investigated further. Results of the present study strongly suggest that the observed decline of the green crab population at Clarke Head was at least partially due to this event. There are important demographic and ecological implications to other benthic species in the Bay of Fundy-Gulf of Maine region if they have been or are still being impacted similarly to this green crab population by an oceanographic event in or beginning in 2013. If invasive species are indeed more vulnerable to climatic changes than native ones (e.g., *Kienzle, 2015*; *MacDonald et al., 2018*), then these observations of green crab declines should serve as a warning of potential changes to other species in the study region, such as native rock crabs or American lobsters. Continued monitoring of this population and further study of other species will be needed to confirm the causes of these observed shifts, their impacts, and whether populations will continue to be negatively impacted.

## ACKNOWLEDGEMENTS

The author thanks the many individuals that assisted in field sampling and contributed information from local recreational fishing at Clarke Head, NS, especially J. Hardy Best, Kelly Quinn of Parrsboro, NS. Jeff Clements, Krystal Woodard, Heather Hunt, and Rémy Rochette of the University of New Brunswick and Joël Chassé of Fisheries and Oceans Canada are also thanked for information about cunners, clams, and lobster settlement and dispersal.

### Funding
The author received no funding for this work.

### Competing Interests
The author declares there are no competing interests.

### Author Contributions
- Brady K. Quinn conceived and designed the experiments, performed the experiments, analyzed the data, contributed reagents/materials/analysis tools, prepared figures and/or tables, authored or reviewed drafts of the paper, approved the final draft.

### Data Availability
The raw data are provided in a Supplemental File.

## Supplemental Information

Supplemental information for this article can be found online at http://dx.doi.org/10.7717/peerj.5566#supplemental-information.

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
