# Peer review of "Dramatic decline and limited recovery of a green crab (Carcinus maenas) population in the Minas Basin, Canada after the summer of 2013"

_PeerJ, doi:10.7717/peerj.5566_

## Round 0.1 · original submission · Major Revisions

The manuscript identifies a major ecological change in 2013 for the green crab and thus should be of high interest to readers. However, as enumerated by reviewer 2, statistical issues must first be addressed to support the conclusions stated in the manuscript.

Reviewer 1 ·

Basic reporting

This paper is extremely well written from a technical perspective with the author using very clear language throughout. There are no grammar issues, and the readability level is high. Literature is well-sourced, with a selection of the most current and seminal references on the topic of green crabs being cited. The figures and tables are well formatted, and the raw data, once re-analysed, confirm the results. The paper follows logically from hypotheses to relevant results and discussion.

Experimental design

I feel strongly that this research fits within the scope of PeerJ’s mandate. The research is topical, and with the regional population crash of 2013, is quite important to the region as it remains an understudied phenomenon. Having been involved in similar long-term studies investigation population fluctuations for this species, I can say definitely that this represents the most thorough attempt at describing the trends observed across the region.

I do however, question the suitability of using shoreline quadrat data as an appropriate methodology for assessing green crab populations in lieu of using more traditional trapping or aquatic survey data. Without traditional indices such as Fukui trapping CPUE, it is difficult to superimpose these results across the region. An more thorough emphasis on the potential drawbacks of using shoreline data should also be included. For example, while the author notes that green crabs may retreat deeper into the intertidal zone, the actual number of crabs that actually do complete this behaviour is unknown and could depend regionally due to many factors – one of which, tidal fluctuations, is quite important in the area sampled. This impact should not be understated. WE do not know how much of the population remains on shore between tidal flushes, and this ratio or number is essential in discussion overall population trends. While the author does include data collected from flounder fishers as an attempt to validate or control for the shoreline data, the potential for this fishing methodology to capture green crabs is not thoroughly explained. If the author were to elaborate on these methodologies, as well as to explain why trapping was not possible, I think the methodology explanation would be improved greatly.

Methodology descriptions aside, I feel that the investigation of these data was of a very high technical standard, with all appropriate statistical analyses performed. However, the author might consider a more robust multivariate methodology for the investigation of the potential biological characteristics that may affect the densities of crabs.

Validity of the findings

This paper presents its findings in a conclusive manner. The conclusions are well stated, and the derivations of said conclusions are not without merit. I find this paper to be a valuable addition to the green crab literature.

Additional comments

I enjoyed reviewing this paper as it attempts to answer a question that I have heard repeated around the region in regards to the population trends over the past few decades. While I do have reservations about the use of shoreline data for inferring population trends, I understand that trapping is not always feasible. However, I am confident that the comparison from year to year, at least for the Minas Basin, is entirely appropriate.

It would be great if you could find some water temperature data for the area. While air data are appropriate to use as a proxy, they tend to fluctuate much more broadly, and may represent a potential issue.

Reviewer 2 ·

Basic reporting

This study is observational, with no experimental component, but the length of time that the study spanned makes it an interesting contribution, since ecological mechanisms can only be studied after patterns are carefully described. However, the data analyses are weak in a number of ways, which requires further work. Detailed comments follow below:
L119. Adding habitat photographs (never a limitation for online articles) would enhance the description of the studied habitats.
L120. This study was done in Clarke Head. How general are the patterns in this entire region? Is Clarke Head an anomaly? Or a good representative of the generality?
L160. At one same site on the Maine coast, I have seen no green crabs during the day, but many during the night. How different would results have been had the author made the surveys at night?
L168. What range of intertidal elevations (in meters above chart datum) were sampled? Intertidal species abundance and size depends heavily on intertidal elevation (see, for example, Scrosati, Grant & Brewster 2012. Density and size gradients across species distribution ranges: testing predictions from the abundant-centre model using the vertical distribution of intertidal barnacles. Vie et Milieu - Life and Environment 62: 197-202).
L197-198. In online articles, where space is not a limitation, there is no need to state “results not shown”. Results should be shown in one way or another. Otherwise, readers are being asked to believe in the text solely based on faith, which is not evidence-based science.
L219. This question seems particularly well suited to be analyzed through multiple linear (or nonlinear) regression, as opposed to separate correlation analyses for each independent variable. Models including all possible combinations of variables could then be compared using their AIC scores, in that way one being able to discern what variables were the most important.
L232-233. Could the specific months that affect larvae, juveniles, adults, etc, be selected on a case-by-case basis, instead of only using fixed “seasonal” groups of months? That approach could increase the chance of detecting significant relationships.
L240. Seawater temperature, air temperature, or both? Intertidal organisms are affected by seawater temperature during high tides and air temperature during low tides. This is addressed later in the text, but it should be explained here, as it is the first mention of temperature.
L261-263. Those values of air temperature were likely not unequivocally related to intertidal organisms, because only values of air temperature calculated during low-tide periods should be used. Matching the timing of low tides to air temperature values should therefore be done in order to use only the values important for intertidal organisms. Values of air temperature calculated during periods of high tide (for the sampled elevations) should be excluded from analyses concerned with air temperature as an ecological factor.
L270. That assumption is not reasonable, because crabs normally stay under macroalgal canopies during low tides to avoid heat stress. Air temperature values, therefore, will usually not tell what crabs are actually experiencing at low tide.
L282-283. This could be approached by comparing models using AIC, as I mention above.
Table 3 must provide a P value for each r coefficient. This is a cornerstone concept in statistics. There will always be an r value for any X-Y relationship, but concluding whether it matters or not can only be done after calculating its significance. In other words, speaking about “high” or “low” values of correlation without making the corresponding significance tests is meaningless.
In Table 3, it is also unclear what “SD”, “Min”, and “Max” represent.
L423. Again, such a conclusion can only be provided when any given r value is found to be significant (usually P < 0.05) after the corresponding statistical test.
L466. Chignecto
As this study is entirely observational, it is not possible to identify causal factors for the observed patterns. Therefore, the Discussion section can only offer speculations on why things happened the way they did. However, the provided speculations, although open to interpretation, are generally reasonable. Overall, because this paper identifies a major ecological change in 2013, it represents a useful contribution, although the statistical issues identified above must first be addressed to allow this paper to be based on sound data analyses.

Experimental design

See the "Basic reporting" section above.

Validity of the findings

See the "Basic reporting" section above.

Additional comments

See the "Basic reporting" section above.

---

## Round 0.2 · accepted · Accept

Thank you for the revised manuscript addressing the points raised by reviewers. Congratulations! Your manuscript is accepted for publication in PeerJ.

# Reviewer 1 ·

Basic reporting

I find that the author has addressed the concerns laid out in the previous review adequately. I find no other issues with this submission. I find it a valuable piece that contributes to Green crab and invasive species research both regionally, and locally.

Experimental design

No comment.

Validity of the findings

No comment.